# DEEP GEOMETRICAL GRAPH CLASSIFICATION

## ABSTRACT

Most of the existing Graph Neural Networks (GNNs) are the mere extension of the Convolutional Neural Networks (CNNs) to graphs. Generally, they consist of several steps of message passing between the nodes followed by a global indiscriminate feature pooling function. In many data-sets, however, the nodes are unlabeled or their labels provide no information about the similarity between the nodes and the locations of the nodes in the graph. Accordingly, message passing may not propagate helpful information throughout the graph. We show that this conventional approach can fail to learn to perform even simple graph classification tasks. We alleviate this serious shortcoming of the GNNs by making them a two step method. In the first of the proposed approach, a graph embedding algorithm is utilized to obtain a continuous feature vector for each node of the graph. The embedding algorithm represents the graph as a point-cloud in the embedding space. In the second step, the GNN is applied to the point-cloud representation of the graph provided by the embedding method. The GNN learns to perform the given task by inferring the topological structure of the graph encoded in the spatial distribution of the embedded vectors. In addition, we extend the proposed approach to the graph clustering problem and an architecture for graph clustering is presented. Moreover, the spatial representation of the graph is utilized to design a new graph pooling algorithm. We turn the problem of graph down-sampling into a column sampling problem, i.e., the sampling algorithm selects a subset of the nodes whose feature vectors preserve the spatial distribution of all the feature vectors. We apply the proposed approach to several popular benchmark data-sets and it is shown that the proposed geometrical approach strongly improves the state-of-the-art result for several data-sets. For instance, for the PTC data-set, the state-of-the-art result is improved for more than 22 %.

## 1 INTRODUCTION

Many of the modern data/signals are naturally represented by graphs (Bronstein et al., 2017; Dadaneh & Qian, 2016; Cook & Holder, 2006; Qi et al., 2017b) and it is of great interest to design data analysis algorithms which can work directly with graphs. Due to the remarkable success of deep learning based methods in many machine learning applications, it is an attractive research problem to design new neural network architectures which can make the deep networks able to work with graphs. The adjacency matrix of a graph exhibits the local connectivity of the nodes. Thus, it is straightforward to extend the local feature aggregation in the Convolutional Neural Networks (CNNs) to the Graph Neural Networks (GNNs) (Simonovsky & Komodakis, 2017; Zhang et al., 2018; Atwood & Towsley, 2016; Niepert et al., 2016; Bruna et al., 2013; Fey et al., 2018). The local feature aggregation in the graphs is equivalent to message passing between the nodes (Zhang et al., 2018; Gilmer et al., 2017). However, if the graph is not labeled (the nodes/edges do not have feature vectors) or the feature vectors of the nodes do not carry information about the similarities between the nodes or any information about their structural role, the message passing may not propagate informative features throughout the graph. Thus, the GNN can fail to infer the topological structure of the graph. Another limitation of the current GNN architectures is that they are mostly unable to do the hierarchical feature learning employed in the CNNs (Krizhevsky et al., 2012; He et al., 2016). The main reason is that graphs lack the tensor representation and it is hard to measure how accurate a subset of nodes represent the topological structure of the given graph. In this paper, we address these two shortcomings of the GNNs.

In the proposed approach, we provide a spatial representation of the graph to the GNN. The spatial representation makes the network aware of the similarity/difference between the nodes and also the location of the nodes in the graph. Accordingly, the local feature aggregation can propagate informative messages throughout the graph. In addition, we propose a down-sampling method to perform graph pooling. The proposed node sampling approach measures the similarities between the nodes in the spatial domain and merges the closest pairs of nodes. The main contributions of the proposed approach can be summarized as follows:

- It is shown that the existing GNNs lack the important embedding step. The proposed approach enhances the capability of the GNNs in inferring the topological structure of the graphs.
- A new graph pooling method is proposed which can be implemented in any GNN. The proposed pooling method preserves the spatial representation of the graph by merging the closest pairs of nodes in the spatial domain.
- To the best of our knowledge, the proposed approach advances the state-of-the-art results for 5 established benchmark data-sets. For instance, the improvement in accuracy with the PTC data-set is more than 22 %.
- An architecture for graph clustering is presented. In contrary to the conventional GNNs which can not be trained to cluster unlabeled graphs, the proposed approach utilizes the geometrical representation of the graph to infer the structure of the graph.

## 1.1 NOTATION

We use bold-face upper-case letters to denote matrices and bold-face lower-case letters to denote vectors. Given a vector $\mathbf{x}$, $\|\mathbf{x}\|_p$ denotes its $\ell_p$ Euclidean norm, and $\mathbf{x}(i)$ denotes its $i^{\text{th}}$ element. Given a set $\mathcal{A}$, $\mathcal{A}[i]$ denotes the $i^{\text{th}}$ element of the set. Given a matrix $\mathbf{X}$, $\mathbf{x}_i$ denotes the $i^{\text{th}}$ row of $\mathbf{X}$, $\mathbf{x}^i$ denotes the $i^{\text{th}}$ column of $\mathbf{X}$, and $\mathbf{X}(i,j) = \mathbf{x}_i(j)$. A graph with $n$ nodes is represented by two matrices $\mathbf{A} \in \mathbb{R}^{n \times n}$ and $\mathbf{X} \in \mathbb{R}^{n \times d_1}$, where $\mathbf{A}$ is the adjacency matrix, $\mathbf{X}$ is the matrix of feature vectors of the nodes, and $d_1$ is the dimension of the attributes of the nodes. The matrix of feature vectors at the input of the $l^{\text{th}}$ layer of the network is denoted by $\mathbf{X}^l \in \mathbb{R}^{n^l \times d^l}$, where $n^l$ is the size of the graph at the output of the $(l-1)^{\text{th}}$ layer of the network and $d^l$ is the length of the feature vectors at the output of the $(l-1)^{\text{th}}$ layer of the network ($\mathbf{X}^0 = \mathbf{X}$ and $n^0 = n$). The set $\mathcal{I}_i$ contains the indices of the neighbouring nodes of the $i^{\text{th}}$ node. We also include $i$ in $\mathcal{I}_i$. The operation $\mathbf{A} \Leftarrow \mathbf{B}$ means that the content of $\mathbf{A}$ is set equal to the content of $\mathbf{B}$. The function $\lfloor y \rfloor$ returns the closest integer which is smaller than or equal to the real number $y$.

## 2 RELATED WORK

The proposed approach translates the graph analysis problem into a point-cloud (Qi et al., 2017a) analysis problem. In the proposed method, the deep network is given the geometrical representation of the graph which is obtained by a graph embedding algorithm. Therefore, in this section we review some of the related research works in graph classification and graph embedding.

*Graph Embedding:* A graph embedding method aims at finding a continuous feature vector for each node of the graph such that the topological structure of the graph is encoded in the spatial distribution of the feature vectors. The nodes which are close on the graph or they share similar structural role are mapped to nearby points by the embedding method. The graph embedding methods such as DeepWlak (Perozzi et al., 2014) and Node2Vec (Grover & Leskovec, 2016) generalize the recent advances in word embedding to graphs. For instance, DeepWalk uses the local information obtained from the random walks to learn the nodes representations by treating the random walks as sentences. We refer the reader to (Ivanov & Burnaev, 2018; Perozzi et al., 2014; Grover & Leskovec, 2016) and the references therein for a more comprehensive review of graph embedding.

*Kernel Based Graph Classification Methods:* Graph kernels are graph analysis tools which are able to measure the similarity between two graphs. They make the kernel machines such as SVM able to work directly on graphs. The primary idea, proposed in (Haussler, 1999), is to decompose a graph into small substructures and to compute the similarity between two graph by adding up the pair-wise similarities between these components. The main difference between the graph kernels is in

their choice of the substructures. The substructures include walks (Vishwanathan et al., 2010), sub-graphs (Kriege & Mutzel, 2012), paths (Borgwardt & Kriegel, 2005), and sub-trees (Shervashidze et al., 2011). Despite the success of graph kernels, they have two major drawbacks. First, the graph kernels are computationally expensive. These methods need to fill out the kernel matrix by computing the similarity between every two graphs in the training data-set. Thus, the complexity of training a kernel based method scales with the square of the size of the training data. In addition, computing each element of the kernel matrix can be computationally expensive. For instance, the computation complexity of the shortest path graph kernel scales with the square of the number of nodes. This high computation complexity makes the algorithm inapplicable to large graphs. In Section 7.2, we compare the performance of the proposed method with some of the graph kernels. Although the computation complexity of the proposed approach can be linear with the number of nodes and the complexity of the training of the proposed method is linear with the size of the training data, it strongly outperforms them on most of the data-sets. The second problem with graph kernels is that the features that they use for classification is independent from the data-set. In contrary to the deep learning based methods, the extracted features are not data driven.

*Graph Neural Networks:* In recent years, there has been a surge of interest in developing deep network architectures which can work with graphs. The main trend is to adopt the structure of the convolutional networks. In contrary to images, we do not have any meaningful order of the neighbouring nodes in graphs. Thus, we can not assign a specific weight to each neighbouring node of a given node. Accordingly, the feature vector of a given node is uniformly aggregated with the feature vectors of its neighbouring nodes. Specifically, the matrix of feature vectors $\mathbf{X}$ is updated as $\mathbf{X} \Leftarrow f(\mathbf{AXW})$, where $\mathbf{W}$ is the weights matrix assigned to all the nodes and $f(\cdot)$ is the element-wise non-linear function. If this operation is repeated $k$-times, a given node receives "messages" from all the nodes with distance up to $k$ steps away (Gilmer et al., 2017).

In addition to the lack of tensor representation in graphs, it is not straightforward to perform pooling on graphs. Thus, the standard approach in the GNNs is to globally aggregate all the feature vectors of the nodes using the element-wise mean function or the element-wise max function. In order to avoid the aggregation over the whole graph and to "keep more vertex information", the recent architecture presented in (Zhang et al., 2018) builds a sequence of a set of sampled nodes and applies a one dimensional CNN to the sequence of the nodes. However, it is evident that a one dimensional space do not have the capacity to represent the structure of a graph. For instance, we lose all the spatial information in an image if we sort the pixels (or their feature vectors) in a 1-dimensional array. In (Ying et al., 2018), a soft graph pooling method was proposed. The pooling method proposed in (Ying et al., 2018) maps the nodes with similar adjacency vectors to the same cluster in the pooled graph. However, large graphs are mostly sparsely conneted. Thus, the adjacency vectors of two nodes which lie in the same cluster can be orthogonal. Therefore, the pooling method presented in (Ying et al., 2018) might not be applicable to large graphs. In addition, the soft graph pooling method does not preserve the sparsity of the adjacency matrix. In (Defferrard et al., 2016; Fey et al., 2018), the graph clustering algorithms were combined with the GNN to perform the graph down-sizing. In this paper, we do not assume that the given graph is clustered and the graph pooling is performed dynamically with respect to the distribution of the learned feature vectors.

## 3 MOTIVATIONS

Consider a 2-dimensional image. This image is represented using a 3-dimensional tensor. A given pixel in the image is surrounded by 8 pixels. Each neighbouring pixel has a distinct relative position with respect to the given pixel. This spatial order makes the CNN able to aggregate the local feature vectors using different weight matrices $\{\mathbf{W}_i\}_{i=0}^8$. Thus, if $\mathbf{x}_0$ represents the feature vector of the center pixel and $\{\mathbf{x}_k\}_{i=1}^8$ represent the feature vectors of the surrounding pixels, the feature vector of the center pixel is updated as $\mathbf{x}_0 \Leftarrow f\left(\sum_{i=0}^8 \mathbf{W}_k \mathbf{x}_k\right)$, where $f(\cdot)$ is the element-wise non-linear function (Goodfellow et al., 2016). The adjacency matrix of a graph exhibits the local connectivities of the nodes. In contrary to the convolution operation applied to images, we can not discriminate between the neighbouring nodes. Accordingly, in the GNNs the feature vector of a given node is updated as

$$\mathbf{x}_0 \Leftarrow f\left(\mathbf{W} \sum_{i=0}^k \mathbf{x}_k\right),\tag{1}$$

where $k$ is the number of neighbouring nodes and $\mathbf{W}$ is the weight matrix assigned to all the nodes. Suppose the graph is not labeled (neither the node nor the edges are labeled). Thus, the feature aggregation (1) only calculates the degree of the given node. In the subsequent message passing, the information about the degree of the nodes is propagated around the graph. There is no mechanism in this approach to learn the topological structure of the graph.

For instance, suppose we have a data-set of clustered unlabeled graphs (by unlabeled here we mean the nodes and the edges are not labeled) and assume that the clusters are not topologically different (a common generator created all the clusters). One class of the graphs consist of two clusters and the other class of graphs consist of three clusters. Consider the simplest case in which there is no connection between the clusters. In addition, assume that we use a typical GNN in which the feature propagation (1) is performed multiple times and all the feature vectors are aggregated using an indiscriminate aggregate function such as the element-wise max function or the element-wise mean function. Suppose that a given graph belongs to the first class, i.e., it consists of two clusters. Define $\mathbf{v}_1$ and $\mathbf{v}_2$ as the aggregated feature vectors corresponding to the first and the second clusters, respectively (the global feature vector of this graph is equal to the element-wise mean/max of $\mathbf{v}_1$ and $\mathbf{v}_2$). Clearly, $\mathbf{v}_2$ can be indistinguishable from $\mathbf{v}_1$ since the clusters are generated using the same generator. Therefore, the feature vector of the whole graph can also be indistinguishable from $\mathbf{v}_1$. The same argument is also true for a graph with three clusters. Accordingly, the representation obtained by the GNN is unable to distinguish the two classes. This example shows that the typical GNN can be unable to learn to solve even a simple graph classification problem. In Section 7.1, we provide three numerical examples to show this phenomena.

An important step in almost all the deep learning based language modeling methods is the embedding step which projects the words with similar meaning or with similar role to close data points in the continuous embedding space (Goldberg & Levy, 2014). Therefore, the next layers of the network learn to solve the given task by analyzing the distribution of the points generated by the embedding layer and also their order in the sequence. In the GNNs, this important embedding layer is missing. If the nodes are not labeled, the network may not understand the difference/similarity between the nodes and the location of the nodes in the graph.

In text data, each sentence is a sequence of words. Thus, if the sentence is represented using a graph, it is always a sequence of nodes where each node is connected to the next node. In addition, there is a fixed dictionary of words. Thus, the nodes of all the graphs which represent the sentences are sampled from a fixed dictionary of nodes. Neither of these two privileges are available in graphs. The structure of the graphs in a data-set are not necessarily the same and we generality cannot assume that all the graphs are built from a fixed dictionary of nodes. Therefore, in the GNN we basically can not have a fixed embedding layer as in the deep networks designed for text data. Therefore, in this paper we use a two step method. The first step is graph embedding to obtain a representation of the graph in the continuous space. The second stage is a neural network which learns to perform the given task by analyzing the spatial representation of the graph. In other word, the proposed approach translates the graph analysis problem into a point-cloud analysis problem.

The second main motivation of the presented work is the lack of a pooling function in most of the existing GNNs (Gilmer et al., 2017; Hamilton et al., 2017; Dai et al., 2016; Duvenaud et al., 2015; Li et al., 2015). Using a pooling function, the GNNs would be able to learn hierarchical features. If the distribution of the feature vectors of the nodes represent the topological structure of the graph, the node sampling problem can be translated into a column/feature sampling problem (Halko et al., 2011; Deshpande & Rademacher, 2010). We use this idea to design a new graph pooling method.

## 4 PROPOSED APPROACH

Inspired by the success of deep networks in analyzing point-cloud data (Qi et al., 2017a;b) and text data, we utilize an embedding step to provide a geometrical representation of the graph to the deep network. The embedding step turns the graph into a point-cloud in the embedding space. The nearby nodes or the nodes which have similar structural role are represented by close points in the embedding space. According to the discussion in Section 3, we cannot include a fixed embedding stage which can handle all the graphs in the given data-set. Thus, we use a graph embedding method to solve the embedding problem for each graph independently. The graph embedding method obtains a representation of the graph in the continuous embedding space (in all the presented experiments,

we use DeepWalk as the graph embedding algorithm). Accordingly, the input to the deep network is $\mathbf{X} \in \mathbb{R}^{n \times (d_1+d_2)}$ which is the concatenation of the embedding vectors and the given nodes attributes. The parameter $d_2$ is the dimension of the embedding space used by the graph embedding algorithm. In this paper, we use a simple GNN and we equip it with our pooling method. We build our network using the following computation blocks:

*Initial Feature Transformation*: This stage is composed of several fully connected layers. Each layer is composed of a linear transformation followed by Batch Normalization (Ioffe & Szegedy, 2015) and an element-wise non-linear function.

*Local Features Aggregation*: Similar to the existing GNNs, this unit combines the feature vector of a node with its neighbouring nodes. The feature vector of the $i^{\text{th}}$ node in the $l^{\text{th}}$ layer of the network is updated as

$$\mathbf{x}_i^l \Leftarrow g\left(\{\mathbf{W}^l \mathbf{x}_k^l\}_{k \in \mathcal{I}_i}\right) \ , \tag{2}$$

where $\mathbf{W}^l$ is the weight matrix assigned to all the nodes. The function $g$ is the aggregate function. In this paper, we use the element-wise $\max$ function.

*Graph Pooling*: One of the main challenges of extending the architecture of the CNNs to graphs is to define a pooling function applicable to graphs. It is hard to measure how accurate a sub-sampled graph represents the topological structure of a given graph. In contrary, it is straightforward to measure how accurate a subset of data points represent the spatial distribution of a set of data points. Since the distribution of the feature vectors of the nodes represents the topological structure of the graph, we define the primary aim of the proposed pooling function to preserve the spatial distribution of the feature vectors. The proposed pooling function down-samples the graph by a factor $2^z$ where $z$ is an integer greater than or equal to 1. In this paper, we always set $z = 1$, i.e., each layer of graph pooling down-sizes the graph by a factor of 2. The proposed pooling function is detailed in Section 4.1. In contrary to the pooling layer used in the CNNs, the way the proposed method down-sizes the graph is not predefined and it depends on the spatial distribution of the feature vectors in each layer of the network. The down-sizing is performed dynamically with respect to the last distribution of the feature vectors.

*Final Aggregator*: After $k$ steps of graph pooling, the given graph is down-sized to a graph of size $n/2^k$. In this paper, we use the element-wise $\max$ function to summarize all the feature vectors into a global representation of the graph.

Figure 1 shows the architecture of the network we used in the presented numerical experiments.

**Remark 1.** *If in the given data edge attributes are available, the edge attributes can be used in the feature aggregation (2). For instance, if $\mathbf{y}_{i,k}$ is the attribute of the edge between the $i^{th}$ node and the $k^{th}$ node, we can concatenate $\mathbf{y}_{i,k}$ with $\mathbf{x}_k$ in (2) to use the edge features.*

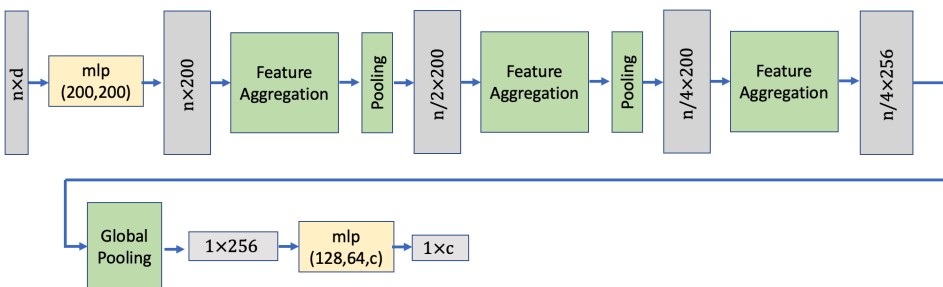

Figure 1: The architecture of the network we used in the numerical experiments. mlp stands for multi-layer perceptron, numbers in bracket are layer sizes. Batchnorm is used for all layers with ReLU. Dropout layers are used for the last mlp.

## 4.1 Adaptive Graph Pooling

Although, it is hard to define a metric to measure how accurate a sub-set of the nodes preserve the topological structure of the graph, it is straightforward to define a metric to measure how accurate

a subset of points represent a set of points. The problem of sampling a sub-set of informative data points is a well-known problem in data analysis. It is mostly known as the column/row sampling problem (Halko et al., 2011; Deshpande & Rademacher, 2010; Drineas et al., 2006). However, most of the column sampling algorithms focus on finding a sub-set of the columns whose span yields an accurate low rank approximation of the given data.

In our application (graph down-sampling), instead of the row space of the data we care about the spatial distribution of the feature vectors. The spatial distribution inherently exhibits the topological structure of the data. The Table of Algorithm 1 presents the proposed node/row sampling algorithm. In the proposed method, first the distance between the data points are computed. Subsequently, the closest pairs of points are found consecutively. Once the sets $\mathcal{A}$ and $\mathcal{B}$ are computed, the nodes in $\mathcal{A}$ are merged with the corresponding nodes in $\mathcal{B}$. In each graph pooling step, if we run Algorithm 1 $z$ time, the graph is down-sampled by a factor of $2^z$. In this paper, we set $z = 1$, i.e., each pooling layer down-sizes the graph by a factor of 2.

As an example, suppose the given graph has 6 nodes $\{a_i\}_{i=1}^6$. The right image of Figure 2 shows the distribution of the nodes in the feature space. The closest pair of nodes is $(a_1, a_2)$. Thus, they are merged and they are represented using a single node in the pooled graph. If we ignore $a_1$ and $a_2$, then $(a_5, a_6)$ is the closest pair of nodes. Similar to $(a_1, a_2)$, $(a_5, a_6)$ are merged and they are represented using a single node in the pooled graph. At the end, the same role is applied to $(a_3, a_4)$. If the graph has an odd number of nodes, the remainig node is merged with itself.

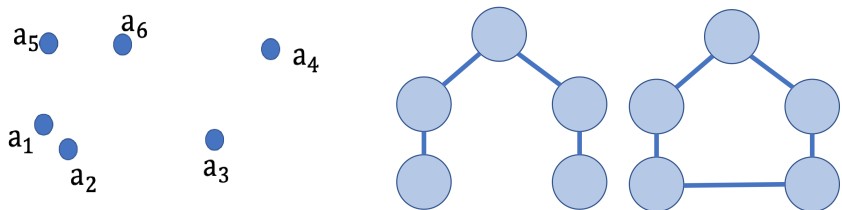

Figure 2: **Left**: The distribution of the nodes of a graph with 6 nodes in the feature space. **Right:** This figure represents two graphs. The circles represent clusters (not nodes). Each cluster contains many nodes. The clusters in the right graph form a loop but in the left graph they do not.

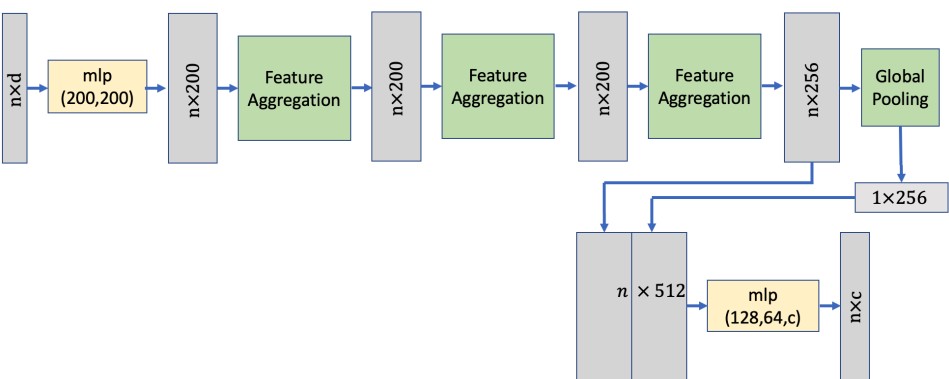

Figure 3: The architecture of the network we used in the numerical experiments for graph clustering. The feature vector of each node after the third Feature Aggregation is concatenated with the global feature obtained with the Global Pooling layer. Therefore, the network is aware of the role of each node in the global structure of the graph. The last mlp block classifies all the nodes independently.

---

**Algorithm 1** Graph Down Sampling

---

**Initialization:** Define empty sets $\mathcal{A}$ and $\mathcal{B}$. Each node in $\mathcal{A}$ is merged with the corresponding node in $\mathcal{B}$.

**Input:** The matrix of feature vectors $\mathbf{X}^l \in \mathbb{R}^{n^l \times d^l}$ in the $l^{th}$ layer of the network.

**1. Compute the $\ell_2$-norms:** Compute vector $\mathbf{q} \in \mathbb{R}^{n^l \times 1}$ such that $\mathbf{q}(k) = \|\mathbf{x}_i^l\|_2^2$. Define matrix $\mathbf{Q} \in \mathbb{R}^{n^l \times n^l}$ such that all the columns of $\mathbf{Q}$ are equal to $\mathbf{q}$.

**2. Compute the distances:** Compute $\mathbf{D} = \mathbf{Q} + \mathbf{Q}^T - 2\mathbf{X}^l\mathbf{X}^{l^T}$. $\mathbf{D}(i,j)$ is equal to the square of the distance between the $i^{th}$ and the $j^{th}$ row of $\mathbf{X}^l$.

**3. Finding the closest pair of nodes:**

**3.1** For $k$ from 1 to $\lfloor \frac{n^l}{2} \rfloor$

**3.1.1** Define $(i,j) = \arg\min_{i',j'} \mathbf{D}(i',j')$.

**3.1.2** Append $i$ to $\mathcal{A}$ and $j$ to $\mathcal{B}$.

**3.1.2** Set $\mathbf{d}_i = \infty, \mathbf{d}^i = \infty, \mathbf{d}_j = \infty, \mathbf{d}^j = \infty$ (Setting these columns/rows equal to infinity means setting them equal to a big number not to sample these columns/rows in the next iterations).

**3.1 End For**

**3.2** If $n^l$ is an odd number, append the index of the remaining node to both set $\mathcal{A}$ and set $\mathcal{B}$.

**4. Merging the pairs of nodes:** The feature vector of the $i^{th}$ node in the down-sampled graph is equal to $\max(\mathbf{x}_{\mathcal{A}[i]}^l, \mathbf{x}_{\mathcal{B}[i]}^l)$. The set of neighbouring nodes of the $i^{th}$ node in the down-sampled graph is the union of the set of neighbouring nodes of the corresponding pair of nodes.

---

## 5 EXTENSION TO GRAPH CLUSTERING

The embedding step turns the graph into a point-cloud. In (Qi et al., 2017a), it was shown that deep neural networks can be successfully trained to perform point-cloud segmentation. The task of graph clustering is similar to point-cloud segmentation. Inspired by the architecture presented in (Qi et al., 2017a), we propose the architecture depicted in Figure 3 for graph clustering. In this architecture, the global feature vector (which is obtained using an element-wise max function) is concatenated with all the feature vectors of the nodes. Thus, the network is aware of the role of each node in the global structure of the graph. Finally, the last layer classifies each node independently. In contrast to data classification, the correct clustering label for each graph is not unique. For each graph with m clusters, there are $m!$ set of correct labels. In each iteration, the loss is calculated using the label which yields the minimum loss.

In this paper, we do not focus on graph clustering and node classification. In one of the presented experiments, we use the proposed architecture to cluster synthesized graphs. We use that experiment to show that the conventional GNNs can not be trained to cluster unlabeled graphs.

## 6 FUTURE WORKS

In the proposed approach, each node of the graph is assigned a distinct data point in the continuous space and close nodes are represented with close data points. Although the graph lacks the tensor representation, we are aware of the positions of the nodes in the continuous space. Thus, inspired by the weighted feature aggregation in the CNNs, if $\mathbf{x}_i$ represents the feature vector of a given node and $\mathbf{x}_k$ is the feature vector of a neighbouring node, we can condition the weight matrix assigned to $\mathbf{x}_k$ on the vector $\mathbf{x}_i - \mathbf{x}_k$. Specifically, define function $\mathbf{H}(\cdot)$ such that its input is a vector and it outputs a weight matrix. Thus, we can aggregate the local features as $\mathbf{x}_i \Leftarrow \sum_{k \in I_i} \mathbf{H}(\mathbf{x}_i - \mathbf{x}_k)\mathbf{x}_k$. In order to build the weight matrix generator $\mathbf{H}(\cdot)$, we can implement a side network (which is trained with the main classifier) to generate the weights for the main network. In (Ying et al., 2018), a similar idea was proposed to condition the weight matrices on the edge attributes. However, an edge attribute may not contain the information which can specify the relation of two neighbouring nodes. In addition, in most of the applications the edge attributes are not available or the graph is sparsely labeled.

# 7 NUMERICAL EXPERIMENTS

In this section, first we provide two experiments with synthetic data to show that the conventional GNNs can fail to infer the topological structure of the graphs. Subsequently, we apply the proposed approach to several established graph classification benchmark data-sets and compare the classification accuracy of the proposed approach against several kernel based methods and multiple recently proposed deep learning based approaches. In all the experiments, we used DeepWalk (Perozzi et al., 2014) to obtain the geometrical representation of the graphs with $d_2 = 12$. We train the network using the implementation of Adam (Kingma & Ba, 2014) method in Pytorch. In the proposed method, a dropout layer with dropout rate 0.6 is used after the last two dense layers. In addition to the performance of the proposed approach with the graph pooling layers, we report the performance of the proposed approach without any graph pooling layer too.

In the proposed approach, the given feature vectors of the nodes are concatenated with their corresponding embedding vectors. Define $c_i \in \mathbb{R}^{d_2 \times 1}$ as the embedding vector of the $i^{\text{th}}$ node of the graph and define the self covariance matrix of the $i^{\text{th}}$ node as $\mathbf{C}^i = \mathbf{c}_i \mathbf{c}_i^T$. If in addition to the embedding vectors we concatenate the vectorized form of the self covariance matrices with the node feature vectors, the feature aggregation block can compute the covariance matrix of the embedding vectors of the connected nodes. Our validations showed that this technique can improve the classification accuracy for some of the data-sets.

## 7.1 SIMPLE GRAPH ANALYSIS TASKS WITH SYNTHETIC DATA

In this section, we define three graph analysis tasks with synthetic data to show that the GNNs without an embedding stage can fail to infer the structure of the graph. We study the performance of the proposed method, the network proposed in (Zhang et al., 2018), and a simple GNN which consists of 4 steps of message passing followed by an aggregate function (element-wise max function) applied to all the nodes. The generated graphs are composed of 3 to 6 clusters (the number of clusters are chosen randomly per graph). Each cluster is composed of 30 to 60 nodes (the number of nodes are chosen randomly per cluster). Each node in a cluster is connected to 5 other nodes in the same cluster. In addition, if we connect two clusters, we make 3 nodes of one cluster densely connected to 3 nodes of the other cluster.

### 7.1.1 DETECTING HIGH LEVEL LOOP

In this task, we trained the classifiers to detect if the clusters in a graph form a loop. Obviously, there are many small loops inside each cluster. Here we trained the algorithms to detect if the clusters form a high level loop. The right image of Figure 2 demonstrates the classification problem. We trained the three networks with 3500 training graphs and tested with 500 graphs. The accuracy of the proposed approach, DGCNN, and the simple GNN are 0.99, 0.55, and 0.53, respectively. It is clear that DGCNN and the simple GNN failed to learn any relevant feature.

### 7.1.2 COUNTING THE NUMBER OF CLUSTERS

We repeated experiment 7.1.1 but we trained the networks to estimate the number of the clusters. Thus, there are 4 classes of graphs. The accuracy of the proposed approach, DGCNN, and the simple GNN are 1, 0.35, 0.33, respectively. The important observation is that the mere extension of the CNNs to graphs can fail to infer the structure of the graph since they are missing the important step of graph embedding.

### 7.1.3 GRAPH CLUSTERING

In this experiment each graph is composed of 3 clusters. The task is to cluster the nodes of a graph into three clusters. We compare the performance of the proposed network depicted in Figure 3 with the deep learning based approach presented in (Kipf & Welling, 2016). The networks were trained with 900 graphs and they were tested with 100 graphs. The average clustering accuracy of the proposed approach and the average clustering accuracy of the method presented in (Kipf & Welling, 2016) are 0.99 and 0.34, respectively. It is evident that (Kipf & Welling, 2016) fails to infer the clustering structure of the graphs. The main reason is that the nodes are not labeled and message passing does not propagate helpful information.

Table 1: Comparison with the kernel based methods.

| Data-set | MUTAG | PTC | PROTEINS | NCI1 | DD B | IMDB |
|---|---|---|---|---|---|---|
| WL | 84.11 | 57.97 | 74.68 | **84.46** | 78.34 | 72.86 |
| RW | 79.17 | 55.91 | 59.57 | $> 3$ days | $> 3$ days | 64.54 |
| GK | 81.39 | 55.65 | 71.39 | 62.49 | 74.38 | 65.87 |
| GEO-DEEP | 93.88 | **76.76** | **79.23** | 74.82 | **80.33** | **73.16** |
| GEO-DEEP (no pooling) | **95.00** | 76.25 | 78.64 | 74.6 | 80.00 | 70.80 |

## 7.2 GRAPH CLASSIFICATION ON REAL DATA

In this section, we study the performance of proposed approach (GEO-DEEP) on several established data-sets (Kersting et al., 2016). Following the conventional settings, we performed 10-fold cross validation, 9 folds for training and 1 fold for testing. Since most of the algorithms can not use the edge attributes, we did not include them in the experiments. We compare the proposed approach (GEO-DEEP) with three graph kernels, the graphlet kernel (GK) (Shervashidze et al., 2009), the random walk kernel (RW) (Vishwanathan et al., 2010), and the Weisfeiler-Lehman sub-tree kernel (WL) (Shervashidze et al., 2011) on six data-sets. Table 1 shows that on most of the data-sets the proposed approach outperforms the kernel based methods which are mostly computationally expensive. In addition, Table 2 compares the proposed approach against 6 recently published deep learning based graph classification methods DGCNN (Zhang et al., 2018), DIFFPOOL (Ying et al., 2018), PSCN (Niepert et al., 2016), DCNN (Atwood & Towsley, 2016), ECC (Simonovsky & Komodakis, 2017), and 2D-CNN (Tixier et al., 2018) on 7 data-sets. The proposed approach outperforms the other methods on 5 data-sets. For instance for the PTC data-set, its accuracy is more than 14 % higher than the previous methods (more than 22 % improvement). In these data-sets, the size of most of the graphs are small. Thus, the proposed approach without the pooling layers achieves close classification accuracy. We conjecture that the pooling layer can yield more clear improvement for large size graphs. Some of the papers which we include their results in Table 2 did not report the variance of the accuracy. The variances of the accuracy of the proposed approach with the real data-sets are 5.2, 4.1, 2.2, 2.5, 1.9, 3, and 3.2 (order similar to the order used in Table 2).

Table 2: Comparison with the other deep learning based approaches.

| Data-set | MUTAG | PTC | PROTEINS | NCI1 | DD | IMDB B | IMDB M |
|---|---|---|---|---|---|---|---|
| DIFFPOOL | - | - | 78.10 | - | **81.15** | - | - |
| ECC | 76.11 | - | - | **76.82** | - | - | - |
| 2D-CNN | - | - | 77.12 | - | - | 70.40 | - |
| DCNN | 0.8013 | 0.5530 | - | 0.6261 | - | - | - |
| DGCNN | 85.83 | 58.59 | 75.54 | 74.44 | 79.37 | 70.03 | 47.83 |
| PSCN | 88.95 | 62.29 | 75.00 | 76.34 | 76.27 | 71.00 | 45.23 |
| GEO-DEEP | 93.88 | **76.76** | **79.23** | 74.82 | 80.33 | **73.16** | 48.13 |
| GEO-DEEP (no pooling) | **95.00** | 76.25 | 78.64 | 74.6 | 80.00 | 70.8 | **49.17** |

## 8 CONCLUSION

We pointed this important fact out that the GNNs which are the mere extension of the CNNs to graphs may not be able to infer the structure of the graphs. In the proposed approach, the GNN analyzes a spatial representation of the graph provided by the embedding step. In other word, the graph data analysis problem is translated into a point-cloud data analysis problem. We also extended the proposed approach to a graph clustering method. In addition, we addressed one of the challenges of extending the architecture of the CNNs to graphs by proposing a graph pooling method. The proposed pooling method merges the closest pairs of nodes in the spatial domain. We showed that

the proposed approach outperforms most of the existing graph classification methods. For instance, for the PTC data-set the proposed approach advances the state-of-the-art result for more than 22 %.

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
