# OpenReview forum: "DEEP GEOMETRICAL GRAPH CLASSIFICATION"
_ICLR.cc/2019/Conference_

### Official Review · AnonReviewer2 · 2018-10-29
**Poorly motivated approach with experimental merits**

**Rating:** 6
**Confidence:** 5

**Review:**

The authors argue that graph neural networks based on the message passing frameworks are not able to infer the topological structure of graphs. Therefore, they propose to use the node embedding features from DeepWalk as (additional) input for the graph convolution. Moreover, a graph pooling operator is proposed, which clusters node pairs in a greedy fashion based on the l2-distances between feature vectors. The proposed methods are evaluated on seven common benchmark datasets and achieve better or comparable results to existing methods. Moreover, the method is evaluated using synthetic toy examples, showing that the proposed extensions help to infer topological structures.

A main point of criticism is that the authors claim that graph convolution is not able to infer the topological structure of a graph when no labels are present. In fact the graph convolution operator is closely related to the Weisfeiler-Lehman heuristic for graph isomorphism testing and can distinguish most graphs in many practical application. Therefore, it is not clear why DeepWalk features would increase the expressive power of graph convolution. It should be stated clearly which structural properties can be distinguished using DeepWalk features, but no with mere graph convolution.
The example on page 4 provides only a weak motivation for the approach: The nodes v_1 and v_2 should be indistinguishable since they are generated using the same generator. Thus, the problem is the mean/max pooling, and not the graph convolution. When using the sum-aggregation and global add pooling, graphs with two clusters and graphs with three clusters are distinguishable again. Further insights how DeepWalk helps to learn more "meaningful" topological features are required to justify its use.

Clustering nodes that are close in feature space for pooling is a reasonable idea. However, this contradicts the intuition of clustering neighboring nodes in the graph. A short discussion of this phenomenon would strengthen the paper in my opinion.

There are several other questions that not been answered adequately in the article.

* The 10-fold cross validation is usually performed using an additional validation set. What kind of stopping criteria has bee use? * It would be helpful to provide standard deviations on these small datasets (although a lot of papers sadly dismiss them).
* I really like the use of synthetic data to show superior expressive power, but I am unsure whether this can be contributed to DeepWalk or the use of the proposed pooling operator (or both). Please divide the results for these toy experiments in "GEO-DEEP" and "GEO-deep no pooling". As far as I understand, node features in different clusters should be indistinguishable from each other (even when using DeepWalk), so I contribute this success to the proposed pooling operator.
* A visualization of the graphs obtained by the proposed pooling operator would be helpful. How do the coarsened graphs look like? Given that any nodes can end up in the same cluster, and the neighborhood is defined to be the union of the neighboring nodes of the node pairs, I guess coarsened graphs are quite dense.
* DiffPool (NIPS 2018, Ying et al.) learns assignment matrices based on a simple GCN model (and thus infers topological structure from message passing). How is the proposed pooling approach related to DiffPool (except that its non-differentiable)? How does it perform when using only the features generated by a GCN? How does it compare to other pooling approaches commonly used, e.g., Graclus? At the moment, it is quite hard to judge the benefits of the proposed pooling operator in comparison to others.


In summary, the paper presents promising experimental results, but lacks a theoretical justification or convincing intuition for the proposed approach. Therefore, at this point I cannot recommend its acceptance.


Minor remarks:

* p2: The definition of "neighbour set" is needless in its current form.
* p2: The discussion of graph kernels neglects the fact that many graph kernels compute feature vectors that can be used with linear SVMs.

-----------
Update:
The comment of the authors clarified some misunderstandings. I now agree that the combination of DeepWalk features and GNNs can encode more/different topological information. I still think that the paper does not make this very clear and does not provide convincing examples. I have update my score accordingly.

---

> ### Author Response · Authors · 2018-11-26
> **Response to reviewer 3**
>
> ٖFirst we would like to thank the reviewer for his/her helpful comments.
>
> --- The motivational example: The problem is not the mean/max aggregator. Even if we use global add pooling, the network can fail to distinguish the clusters. For instance, suppose each node is connected to a certain number of nodes within its cluster but the size of the clusters can vary. In this case it is not possible for the GNN to learn the number of the clusters using graph convolution followed by global add pooling. We actually tried global adding, it did not change the result. Please note even DGCNN completely failed to learn anything.
> In the revision, we have added another synthetic experiment. In this new experiment we show that the GNNs fail to learn to cluster graphs. In contrary, the proposed approach turns the graph into a point-cloud in the embedding space. In the revision, we show how we can use the point-cloud representation to design a graph clustering method.
>
> Our main motivation for employing graph embedding is to transform the graph analysis task to a point-cloud analysis task. With the point-cloud representation, the network is aware of the differences/similarities between the nodes and it is also aware of the location of a node in the graph. This extra information can enhance the ability of the network in inferring the structure of the graph. We supported this idea with experiments on synthetic data and real data.
>
> --- Contradiction between clustering in feature space with clustering nearby nodes: Basically, a graph embedding method (such as DeepWalk) projects nearby nodes to nearby data points in the embedding space. Thus, there is no contradiction between the proposed approach with the method based on clustering nearby nodes.
> In addition, clustering nearby nodes is not a data driven method. There is no rigorous analysis showing that clustering nearby nodes leads to the best performance. In the proposed approach, the pooling is carried out dynamically. Thus, the network learns the pooling operation in a data-driven way. For instance, two nodes can be far from each other on the graph but they could have similar structural role. Thus, the network can learn to merge even the nodes which are far from each other.
>
> ---The success with synthetic data (“I contribute this success to the proposed pooling”): The success of the proposed approach with the synthetic experiment can not be contributed to the pooling layers  because we reported the result of the network which does not use the pooling layers.
> The reason that the proposed approach works perfectly in this case is that graph embedding transforms the graph analysis task into a point-cloud analysis task. Deep networks has been very successful in analyzing point-cloud data [arXiv:1612.00593].
>
> --- Comparing to other pooling methods: We are comparing to two other methods, both published in 2018. They employ new graph pooling operators. One is DiffPool and the other one is DGCNN.
> In DGCNN, the nodes of the graph are sorted. It is evident that sorting in a one dimensional space can destroy the topological structure of the graph. We showed in the synthetic experiments that DGCNN fails to learn to solve the simple graph classification tasks.
>
> In DiffPool the node assignment is learned using a separate GNN. If two nodes have similar adjacency vectors, they will be assigned to the same cluster in the pooled network. However, we think that DiffPool may not be applicable to large graphs because the adjacency matrices of the large graphs are mostly very sparse. Thus, even if two nodes lie in the same cluster, their adjacency vectors can be completely orthogonal. Thus, the network can not figure out which nodes belong to the same cluster. In addition, DiffPool does not preserve the sparsity of the adjacency matrix. It can increase the computation complexity for large graphs.
> In sharp contrast, the proposed approach do not need to process the adjacency matrix. The proposed pooling method aims to preserve the spatial distribution of the feature vectors of the nodes. In addition, we can leverage randomized column sampling techniques to significantly reduce its computation complexity for large graphs.
>
> --- “How does it (pooling method) perform when using only the features generated by a GCN?”: basically, we are able to use the proposed pooling method because the embedding step provides a spatial representation of the graph. The proposed pooling method without the spatial information is not meaningful (It would be similar to downsizing a point-cloud without having the location of the points).
>
> --- Experiments and variances: Some of the papers which we compare with them did not report the variances. In order to have a consistent table, we did not include the information about variances in the table. We have added the information about the variances to the revision.
> For the experiments with real data, we follow the setting used in the previous papers (PSCN, ECC,DGCNN)

---

> > ### Comment · AnonReviewer2 · 2018-11-29
> > **Review updated**
> >
> > Thanks for your reply, this clarified some misunderstandings. I have updated my review accordingly.

---

### Official Review · AnonReviewer1 · 2018-10-31
**Lack of novelty and poorly executed experiments**

**Rating:** 3
**Confidence:** 4

**Review:**

The authors propose a method for learning representations for graphs. The main purpose is the classification of graphs.

The topic is timely and should be of interest to the ICLR community.

The proposed approach consists of four parts:

Initial feature transformation
Local features aggregation
Graph pooling
Final aggregator

Unfortunately, each of the part is poorly explained and/or a method that has already been used before. For instance, the local feature aggregation is more or less identical to a GCN as introduced by Kipf and Welling. There are now numerous flavors of GCNs and the proposed aggregation function in (2) is not novel.

Graph pooling is also a relatively well-established idea and has been investigated in several papers before. The authors should provide more details on their approach and compare it to existing graph pooling approaches.

Neither (1) nor (4) are novel contributions.

The experiments look OK but are not ground-breaking and are not enough to make this paper more than a mere combination of existing methods.

The experiments do not provide standard deviation. Graph classification problems usually exhibit a large variance of the means. Hence, it is well possible that the difference in mean is not statistically significant.

The paper could also benefit from a clearer explanation of the method. The explanation of the core parts (e.g., the graph pooling) are difficult to understand and could be made much clearer.

---

> ### Author Response · Authors · 2018-11-26
> **Response to reviewer 2**
>
> First we would like to thank the reviewer for his/her constructive comments.  We thoroughly edited the paper to address the comments.
>
> The paper has an important message. A GNN without an embedding step can be meaningless. We show this fact with synthetic experiments and the experiments with real data. With synthetic data we show that the GNN basically can not learn to perform the given simple tasks. For some of the real data-sets, the improvement is huge. For instance, for the PTC data-set the improvement is 22 % (62 ==> 76). Moreover, a new architecture based on the proposed idea for graph clustering is proposed.
>
> --- Novelty of the presented architecture: In this paper we do not propose a new convolution layer or a new architecture for the GNNs. We employ a simple GNN depicted in Figure 1.
> The main contribution of the paper is twofold.
> First, we make an important point that the GNNs can fail to infer the structure of the unlabeled  graphs. We show this important point through clear synthetic experiments. The experiments show that convolutional graph neural networks can fail to learn to perform even simple graph classification tasks. We argue that similar to the NLP tasks, a GNN requires an embedding step to make the network aware of the differences/similarities between the nodes of the graph. The embedding step turns the graph analysis task into a point-cloud analysis problem.
> In addition, we showed that although we use a simple GNN, the embedding step can significantly improve the results for many of the real datasets. For instance:
> PTC: 14 % higher accuracy (22 % improvement !)
> MUTAG: 6 % higher accuracy (6 % improvement)
> These are significant improvements while we are employing a simple GNN.
> In addition, we have added a small section to the revision to extend the proposed approach to a graph clustering method. The proposed method leverages the point-cloud representation of the graph. We have added a new experiment which shows that the conventional GNNs cannot be trained to cluster unlabeled graphs.
>
> Our second main contribution is the proposed pooling method. The presented idea is simple. We merge the joint closest nodes in the feature space because the embedding step provides a spatial representation of the graph. In contrast to the previous methods, we do not need to run a graph clustering algorithm. In addition, it can down-sample the graph by a fix predefined factor.  Moreover, in contrast to the soft pooling method, it is applicable to the sparsely connected graphs and the sparsity of the adjacency is not lost.
>
> --- "Graph pooling is also a relatively well-established idea and has been investigated in several papers before. The authors should provide more details on their approach and compare it to existing graph pooling approaches.":
> Yes, pooling is an established idea in processing graphs and point-clouds. In this paper, we propose a novel graph pooling method. In the proposed approach we do not need to run a graph clustering method and the proposed method uses the spatial distribution of the nodes to perform pooling. The proposed method merges the joint nodes which are the closest nodes in the spatial domain. Thus, even if two nodes are far from each other on the graph, they can be merged if they have similar topological roles in the graph.  In this paper, we are comparing the proposed pooling method with two new deep learning based approaches which employed new graph pooling layers (DGCNN and DIFFPOOL).
>
> --- Clarifying the pooling method: In the revised paper, we have edited the pooling method. An example has been added to the revision to explain the proposed method.
>
> ---  "The experiments do not provide standard deviation. Graph classification problems usually exhibit a large variance of the means. Hence, it is well possible that the difference in mean is not statistically significant": Unfortunately some of the deep learning based methods that we are comparing with them did not report the variance. Thus, even if we report the variance, the reader can not make a comparison. We have added the information about the variance of the proposed approach with the real data-sets to the revised paper.
>
> --- "The experiments look OK but are not ground-breaking and are not enough to make this paper more than a mere combination of existing methods.":
> We make an important point in this paper that the GNNs can fail to learn to perform even simple graph analysis Tasks. We provide a clear evidence that message passing without a proper input does not necessarily infer the structure of the graph. In addition, it is shown that with the embedding step, our GNN can significantly advance the state-of-the-art results on the real data-sets. For instance, for the PTC data-set, the improvement is more than 22 %.

---

> > ### Comment · AnonReviewer1 · 2018-12-03
> > **Response**
> >
> > Dear authors,
> > Thank you for your response. I will stick to my initial score. The paper needs a lot more work. Especially the description of the novel contributions (as claimed: pooling etc.) need an overhaul and need to be compared to existing graph pooling methods.

---

> > > ### Author Response · Authors · 2018-12-03
> > > **Response**
> > >
> > > 1 - Thanks for the comment. The paper has been thoroughly edited.
> > >
> > > 2- The novel contributions are explicitly explained in abstract and in Section 1. The main contribution is to translate the graph analysis task into a point-cloud analysis task.
> > >
> > > 3 - The proposed pooling method is compared with existing pooling method (including DGCNN and Diffpool).
> > > In DGCNN, the nodes of the graph are sorted. It is evident that sorting in a one dimensional space can destroy the topological structure of the graph. We showed in the synthetic experiments that DGCNN fails to learn to solve the simple graph classification tasks.
> > >
> > > In DiffPool the node assignment is learned using a separate GNN. If two nodes have similar adjacency vectors, they will be assigned to the same cluster in the pooled network. However, we think that DiffPool may not be applicable to large graphs because the adjacency matrices of the large graphs are mostly very sparse. Thus, even if two nodes lie in the same cluster, their adjacency vectors can be completely orthogonal. Thus, the network can not figure out which nodes belong to the same cluster. In addition, DiffPool does not preserve the sparsity of the adjacency matrix. It can increase the computation complexity for large graphs.
> > > In sharp contrast, the proposed approach do not need to process the adjacency matrix. The proposed pooling method aims to preserve the spatial distribution of the feature vectors of the nodes. In addition, we can leverage randomized column sampling techniques to significantly reduce its computation complexity for large graphs.

---

### Official Review · AnonReviewer3 · 2018-11-02
**A paper addressing an interesting problem, but lacks clarity and hard to understand, tech novelty is unknown**

**Rating:** 4
**Confidence:** 4

**Review:**

This paper proposes a deep GNN network for graph classification problems using their adaptive graph pooling layer. It turns the graph down-sampling problem into a column sampling problem. The approach is applied to several benchmark datasets and achieves good results.

Weakness

1.	This paper is poorly written and hard to follow. There are lots of typos even in the abstract. It should be at least proofread by an English-proficient person before submitted. For example, in the last paragraph before Section 3. “In Ying et al. ……. In Ying et al.”
2.	In paragraph 1 of Section 3, there should be 9 pixels around the center pixel including itself in regular 3x3 convolution layers.
3.	The definition of W in Eq(2) is vague. Is this W shared across all nodes? If so, what’s the difference between this and regular GNN layers except for replacing summation with a max function?
4.	The network proposed in this paper is just a simple CNN. GNN can adopt such kind of architectures as well. And I didn’t get numbers of first block in Figure 1. The input d is 64?
5.	The algorithm described in Algorithm 1 is hard to follow. There are some latex tools for coding the algorithms.
6.	The authors claim that improvements on several datasets are strong. But I think the improvement is not that big. For some datasets, the network without pooling layers even performs better at one dataset. The authors didn’t provide enough analysis on these parts.

Strength:
1.	The idea used in this paper for graph nodes sampling is interesting. But it needs more experimental studies to support this idea.

---

> ### Author Response · Authors · 2018-11-26
> **Response to reviewer 1**
>
> First we would like to thank the reviewer for his/her helpful comments. We thoroughly edited the paper and added new materials.
>
> The paper has an important message. A GNN without an embedding step can be meaningless. We show this fact with synthetic experiments and the experiments with real data. With synthetic data we show that the GNN basically can not learn to perform the given simple tasks. For some of the real data-sets, the improvement is huge. For instance, for the PTC data-set the improvement is 22 % (62 ==> 76). Moreover, a new architecture based on the proposed idea for graph clustering is proposed.
>
> --- Typos:  We thoroughly edited the paper. We have edited those parts which could be vague for the reader. In addition, we have added an example to clarify how the pooling method works.
>
> ---Number of pixels: The reviewer is right about the number of pixels in a 3*3 convolution window. But in that paragraph we are saying that a pixel has 8 neighboring pixels in a 3*3 window. These two statements are not in contradiction.
>
> --- "The definition of W in Eq(2) is vague. Is this W shared across all nodes? If so, what’s the difference between this and regular GNN layers except for replacing summation with a max function?": The weight matrix W is shared across all the nodes.  In this paper we use the conventional graph convolution. We do not propose a new method for graph  convolution.
> Our contribution is twofold.
> First, we make an important point about the GNNs. We argue that similar to the neural network employed for the NLP tasks, GNNs needs an embedding step to make the GNN aware of the difference/similarity between the nodes of the graph. The embedding step turns the graph analysis task into a point-cloud analysis problem.  We support our idea with multiple experiments. We show through clear synthetic experiments that the convolutional graph neural networks can fail to learn to perform even simple graph classification tasks.
> In addition, we show that although we employ a simple GNN, the embedding step can significantly improve the results for many of the real data-sets. For instance:
>
> PTC: 14 % higher accuracy (22 % improvement !)
> MUTAG: 6 % higher accuracy (7 % improvement !)
> These are significant improvements while we are employing a simple GNN.
> Moreover, we have added a small section to the revision which shows that we can use the spatial representation of the graph to design a new graph clustering method (Fig. 3). We have provided a new experiment which shows that the conventional GNN can not be trained to cluster unlabeled graphs. The proposed architecture for graph clustering leverages the point-cloud representation of the graph. It combines the local representation of the nodes with the global representation of the graph to  identify the clusters of the graph.
>
> Our second main contribution is the proposed pooling method. The presented idea is simple. We merge the joint closest nodes in the feature space because the embedding step provides a spatial representation of the graph. In contrast to the previous methods, we do not need to run a graph clustering algorithm. It can down-sample the graph by a fix predefined factor (by a factor of $2^z$ where z can be determined). Moreover, in contrast to the soft pooling method, it is applicable to the sparsely connected graphs and the sparsity of the adjacency is not lost.
>
> --- the mlp block: The first block is a simple multi-layer perceptron. The first layer of the mlp block transforms the input to a 64 dimensional vector. The next layer of the mlp block, transforms the 64 dimensional vector to a 128-dimensional vector and so on. We have slightly changed this block in the revision.
>
> --- The explanation of the pooling method: We have edited the algorithm and we have added an example to the revised paper.
>
> ---"I think the improvement is not that big. For some data-sets, the network without pooling layers even performs better at one dataset. The authors didn’t provide enough analysis on these parts":
> We report the result on many real data-sets which we advance the state-of-the-art for most of them. For some of the data-sets we advance the state-of-the-art result significantly. For instance, we improve the result for the PTC data-set for more than 22 % .
> The network which has the pooling layers yields the state-of-the-art results for some of the data-sets. In the data-sets which we used in our experiment, the size of the graphs are small. Thus, the network which does not has the pooling layers also yields competitive results.
> Similar phenomena was observed in the previous papers too. For instance, when Pointnet++ [arXiv:1706.02413] was proposed to add local feature aggregation to Pointnet Network [arXiv:1612.00593], the results were slightly improved for only few data-sets.  There is no rigorous analysis of deep networks available in the literature. We do not have a clear analysis of the effect of the pooling layers on the performance of the GNNs.

---

### Meta-Review · Area_Chair1 · 2018-12-17
**Scores low on presentation quality, motivation and experimental results**

**Confidence:** 5
**Recommendation:** Reject

**Metareview:**

The extension of convnets to non-Euclidean data is a major theme of research in computer vision and signal processing.  This paper is concerned with Graph structured datasets. The main idea seems to be interesting: to improve graph neural nets by first embedding the graph in a Euclidean space reducing it to a point cloud, and then exploiting the induced topological structure implicit in the point cloud.

However, all reviewers found this paper hard to read and improperly motivated due to poor writing quality. The experimental results are somewhat promising but not completely convincing, and the proposed framework lacks a solid theoretical footing. Hence, the AC cannot recommend acceptance at ICLR-2019.